# Ambient seismic noise analysis of LARGE-N data for mineral exploration in the Central Erzgebirge, Germany

Trond Ryberg[1], Moritz Kirsch[2], Christian Haberland[1], Raimon Tolosana-Delgado[2], Andrea Viezzoli[3], and Richard Gloaguen[2]

[1] Helmholtz Centre Potsdam GFZ German Research Centre for Geosciences, Telegrafenberg, 14473 Potsdam, Germany
[2] Helmholtz-Zentrum Dresden-Rossendorf (HZDR), Helmholtz-Institut Freiberg für Ressourcentechnologie (HIF), Chemnitzer Str. 40, 09599 Freiberg
[3] Aarhus Geophysics, Aps, Voldbjergsvej, Risskov, DK-8200, Denmark

*Correspondence to*: Trond Ryberg (trond@gfz-potsdam.de)

**Abstract.** Ambient seismic noise tomography is a novel, low-impact method to investigate the Earth's structure. While most passive seismic studies focus on structures at crustal scales, there are only few examples of this technique being applied in a mineral exploration context. In this study, we performed an ambient seismic experiment to ascertain the relationship between the shallow shear wave velocity and mineralized zones in the Erzgebirge in Germany, one of the most important metal provinces in Europe. Late Variscan mineralized greisen and veins occurring in the Geyer-Ehrenfriedersdorf mining district of the Central Erzgebirge were mined from medieval times until the end of the 19th century. These occurrences represent a significant resource for commodities of high economic importance, such as tin, tungsten, zinc, indium, bismuth and lithium. Based on ambient noise data from a dense "LARGE-N" network comprising 400 low-power, short-period seismic stations, we applied an innovative tomographic inversion technique based on Bayesian statistics (transdimensional, hierarchical Monte Carlo search with Markov Chains using a Metropolis/Hastings sampler) to derive a three-dimensional shear wave velocity model. An auxiliary 3D airborne time-domain electromagnetic dataset is used to provide additional insight into the subsurface architecture of the area. The velocity model shows distinct anomalies down to approximately 500 m depth that correspond to known geological features of the study area, such as (a) gneiss intercalations in the mica schist-dominated host rock, imaged by a SW-NE striking low velocity zone with a moderately steep northerly dip, and (b) a NW-trending strike-slip fault, imaged as a subvertical linear zone cross-cutting and offsetting this low velocity domain. Similar to the velocity data, the electromagnetic data exhibit north-dipping (high-conductivity) structures in the mica schists, corresponding to the strike and dip of the predominant metamorphic fabric. An unsupervised classification performed on the bivariate 3D dataset yielded nine spatially coherent classes, one of which shows a high correspondence to drilled greisen occurrences in the roof zone of a granite pluton. The relatively high mean shear velocity and resistivity values of this class could be explained by changes in density and composition during greisen formation as observed in other areas of the Erzgebirge. Our study demonstrates the great potential of the cost-efficient and low-impact ambient noise technology for mineral exploration, especially when combined with other independent geophysical datasets.

## 1 Introduction

Passive seismic methods using the ambient seismic noise field have become very popular during the last years for imaging the Earth's subsurface structure. In particular, the discovery of deriving the Green's functions between many seismic stations by data cross correlation turned out to be a game changer in seismology (Campillo & Paul, 2003). These Green's functions correspond to a 'virtual source' located at any given seismic station being recorded at all other stations, thus allowing to infer the velocity structure (and, in turn, the geological structure) without earthquake signals or controlled sources necessary for traditional imaging methods (Bensen et al., 2007). The use of ambient noise fields (from wind, traffic, sea waves etc.) instead of signals from controlled sources (vibroseis, explosions) has several advantages: data acquisition is more environmentally friendly, logistically less demanding, and comparatively cheap. Passive seismic can be applied on a wide range of spatial scales and can yield similar near-surface spatial resolutions as refraction seismic imaging. Recent advances in electronic components made the development of low-power recording systems possible which are used for deployments of large/dense, temporary networks („LARGE-N", formed by hundreds to thousands of nodes). While ambient seismic noise analysis has successfully been used in geodynamic investigations (e.g., Green et al., 2020), shallow site characterization (e.g., Hannemann et al., 2014) and geothermal site characterization (Martins et al., 2020), applications in mineral exploration are relatively sparse (e.g., Hollis et al., 2018; Da Col et al., 2020; Xie et al., 2021; Colombero et al., 2021).

In this paper we describe a passive seismic survey which has been conducted to study technical and methodological aspects of passive seismic methods as well as their usefulness for the application in an exploration context.

A worldwide growing demand in raw materials largely in response to the widespread deployment of green energy technologies, electrification of transport, and digitalization (e.g., Sovacool et al., 2020), has led to renewed exploration activity in the Erzgebirge metallogenic belt (e.g., Dittrich et al., 2020; Burisch et al., 2021), which is well-endowed with high-tech metals such as Sn, W, Zn, In, Ge, Bi, and Li (Štemprok and Seltmann, 1994). The study area in the Geyer-Ehrenfriedersdorf mining district (Central Erzgebirge) was selected given the local occurrence of greisen and vein deposits with significant potential for future ore discoveries. Because of the physico-chemical processes associated with ore formation, ore bodies may differ in their petrophysical properties from the surrounding rocks. Hence, through observation and interpretation of variations in geophysical properties, areas of ore formation can be delineated (Parasnis, 1986; Telford et al., 1992), complementing data from costly field and drilling-based exploration work. The geophysical method used for the delineation of mineral deposits needs to be adapted to the geological setting and type of mineralization. In this case, tin and lithium-bearing greisen and vein deposits similar to those occurring in the Geyer-Ehrenfriedersdorf area have been documented to differ from the surrounding host rocks in terms of their seismic velocities and electrical conductivities (Müller-Huber and Börner, 2017). However, the region is densely populated and contains environmentally sensitive areas, placing additional constraints on the chosen mineral exploration techniques. Ambient seismic and airborne electromagnetics are considered environmentally sensitive and socially acceptable exploration technologies that are well suited in this context. The study is carried out within the framework of the EU-funded INFACT project (2017–2021), which sought to establish

infrastructures and protocols for the trialing of exploration technology. Being located within an INFACT reference site, this study benefits from a large portfolio of exploration data for validation and integration purposes. The results of this study, apart from increasing the robustness of the benchmark targets of the INFACT reference site, will contribute to improving targeting of polymetallic greisen-type mineralization.

## 2 Methods

### 2.1 Study area, petrophysical constraints, and data acquisition

In July/August 2020 we carried out a passive seismic experiment at the INFACT reference site in Geyer-Ehrenfriedersdorf, Erzgebirge, SE-Germany. The study area measures 1.0 x 1.7 km, has an elevation of 570–710 m a.s.l. and is located in a dense forest between the villages of Ehrenfriedersdorf to the SE and Geyer to the SW (Fig. 1). Underneath a thin (on average ca. 3.5 m, locally thickening to 10 m) veneer of Quaternary sediments, the area is underlain by the Geyer granite pluton of late Variscan age (325–318 Ma, Förster and Romer, 2010) emplaced into a suite of Proterozoic to Early Paleozoic metamorphic rocks ranging from high-pressure two-mica schists with muscovite gneiss intercalations in the SE to phyllites in the NW. The Greifensteine, a locally renowned rock formation just outside the study area to the NW (Fig. 1A), forms the exposed top of this granite, and, based on drilling evidence, it is known to drop off steeply due SE to a depth of ca. 400 m below surface in the SE part of the study area (Fig. 1B). The Geyer granite is a medium-grained monzogranite that belongs to a suite of high-F, high-P2O5 Li-mica granites in the Central Erzgebirge (Förster et al., 1999). It is locally associated with porphyritic microgranite dykes. The dominant Late Paleozoic metamorphic fabric trends SW-NE and generally dips NW, with the dip angle increasing from east to west (Hösel et al., 1994). Post-orogenic tectonics led to the formation of a series of steep, NW-SE and NE-SW striking faults including the prominent Greifenbach fault associated with variable lateral offsets of up to 100 m (Fig. 1A).

The Erzgebirge is endowed in world-class magmatic-hydrothermal ore deposits hosted by a variety of vein and metasomatic structures that were extensively mined for silver, tin, bismuth and cobalt since the Bronze Age (Tolksdorf et al., 2019). The most important mineralized structures in the study region are closely associated with the emplacement of the Geyer granite, and include stockwork-like greisen bodies and veins occurring in the endo- and exo-contact of the granite, respectively (Hösel et al., 1994). Greisen are metasomatic granitic rocks composed of quartz, mica and topaz, and are a common host of tin, tungsten and lithium ores in the Erzgebirge (Štemprok and Seltmann 1994, Dittrich et al., 2020). According to Jung and Seifert (1996) the (Sn-W) mineralization occurred in three stages resulting in a zonal distribution with respect to the distance from the granite: (i) a first stage of greisenization is associated with the formation of lithium micas and topaz in the endocontact of the pluton as 2–10 thick peripheral zones and apophyses, (ii) a second stage of greisenization led to the formation of a Sn-W-As paragenesis with cassiterite, wolframite and arsenopyrite as a vein-shaped exo-contact mineralization of 0.1 to 1 m thickness (i.e., Sn-greisen veins in Fig. 1A), and (iii) a late-stage hydrothermal stage involved fluorite and sulfide precipitation. The Sn-W-As succession contains a variety of commodities such as Sn and W and other

metals of economic importance such as In, Cu, Co, Bi, Sb, and Au. Although largely eroded or mined out near surface, exploration drilling in the study area from about 1960 to 1990 has documented remaining mineralized greisen occurrences at depth, with reserves of stringer and greisen ores in the Ehrenfriedersdorf mining district being calculated at 17,100 tons (Hösel et al., 1994). Although the endo-contact greisen mineralization is considered impoverished in terms of Sn-W content (Hösel et al., 1994; Jung and Seifert, 1996), it is known to contain lithium, which, being the dominant battery technology for electric vehicles, is currently in high demand worldwide. Li-Sn-W greisen deposits in the nearby Zinnwald/Cínovec area are currently being developed into an operating mine (Dittrich et al., 2020).

Direct information on surface wave velocity is not available for the rocks of the study area, but petrophysical information from a variety of sources that relate to this parameter and can be used to support the geological interpretation. For instance, bulk density is a parameter critically influencing the seismic velocity in rocks (e.g., Salah et al., 2018). Based on a report by Klemps and Lindner (1985), the two-mica schist in the Geyer-Ehrenfriedersdorf area has a median density of 2.76 g/cm$^3$ (hornfels variety up to 2.80 g/cm$^3$), muscovite gneiss a median of 2.65 g/cm$^3$ and granite a median of 2.62 g/cm$^3$ (Fig. 2A). Greisen in the study area exhibit nominally higher densities (median = 2.69 g/cm$^3$) than the granite. Mineralogical changes induced by greisen alteration of granitic rocks are known to cause an increase in grain density due to the metasomatic replacement of less dense minerals such as albite or K-feldspars by denser minerals such as mica and topaz during greisenization. However, greisenization also induces a porosity increase due to the molar volume decrease associated with mineral replacement reactions (Launay et al., 2019). Density values given in Klemps and Lindner (1985) represent water-saturated density and not bulk density, i.e., they only partially include porosity (from sealed-off or occluded pores), so the bulk density of granite and greisen may not be different if greisen had a higher porosity than granite. Borehole neutron gamma logs from drillholes in the study area (see representative log in Fig. 2B), which are sensitive to the amount of hydrogen atoms in a formation and are commonly used as a proxy for porosity, show no discernable difference between greisen and their host granite (according to a report by Sonntag [1977] summarizing borehole geophysical data from a variety of drill holes in the area). This indicates that the porosity which initially developed in the greisen facies may have been reduced by subsequent mineral precipitation (Launay et al., 2019). Greisen occurrences of similar age and composition from the nearby Altenberg area in the Erzgebirge also exhibit low porosity values (median = 2.63 %; Müller-Huber and Börner, 2017), even lower than their granite host rocks (median = 6.7 %). In terms of electrical resistivity, the primary lithologies two-mica schist, muscovite gneiss and granite cannot be differentiated according to a summary report for borehole geophysical data from a drilling campaign just north of the study area (Tita, 1978). Variations in resistivity are instead attributed to differences in the degree of fracturing, alteration, and mineralization. There is no reference to greisen in this report, but in the nearby Altenberg area, the greisen characteristically exhibits higher specific electrical resistivities (~3250 vs. ~1485 Ωm) compared to their granite host rocks (Müller-Huber and Börner, 2017).

400 seismic stations were deployed to record ambient seismic noise continuously for 10 days. The instruments were installed in an area of approximately 1700 x 1100 m on a dense, regular grid with ~70 m spacing between the grid nodes. We used

DIGOS/Omnirecs digital CUBE recorders equipped with 1- and 3-component seismic sensors (4.5 Hz Eigenfrequency). Accurate timing of the recordings was realized through cycled GPS, recording at 400 samples per second.

## 2.2 Data processing and inversion

Ambient noise techniques have been applied to recover the Green's functions between all stations (80,000) by data cross-correlation (Campillo and Paul, 2003; Wapenaar, 2004; Curtis et al., 2006). As mentioned before, these Green's functions correspond to virtual sources at a given station recorded at all other stations. To derive Green's functions, we applied the data processing to the vertical component noise recordings described in Bensen et al. (2007): it consists mainly of splitting the data records into daily segments, one-bit normalization, spectral whitening and cross-correlation of the traces. The resulting daily cross-correlations are finally stacked. Since the vertical components of all receivers are identical, we did not remove instrument responses. The Green's functions are typically dominated by dispersive surface waves (Rayleigh waves). Before calculating the Green's functions we analysed the characteristics of the ambient noise, i.e. the frequency content and its directional properties. We checked the directional properties of the noise by plain-wave beamforming in the time domain (see e.g. Schweitzer et al., 2012, Harjes & Henger,1973). For the analysis we used a subset of 22 stations and calculated the beam power of 10-minute time windows after bandpass filtering (1.2-4.5Hz, 4.5-10 Hz, 10-20 Hz). Six consecutive beamforming results were stacked to yield hourly beamforming plots. Fig. 3 shows examples of these beamforming plots (normalized beam power) together with a spectrogram (power spectral density; single station) for an arbitrary day (2020-07-31). Noise levels vary significantly with time. Low-frequency noise is often coming from north-north-westerly directions (apparent velocities 2.5-4 km/s), however, also other directions are represented. In the higher-frequency bands the noise is also coming from a broad range of directions and slownesses, however, also some rather constant sources can be identified (e.g. signal around 10 Hz in the spectrogram of Fig. 3 coming from ~NW with an apparent velocity of around 6.6 km/s). The virtual shot gathers constructed from the vertical component recordings are shown in Fig. 4A for all station pairs and Fig. 4B for a single virtual shot gather. The latter gather shows a significant asymmetry of the causal and acausal phases. In the next step the Rayleigh dispersion curves were determined in an interactive way for a large number of cross-correlations (FTAN method from Dziewonski et al, 1969 with band=0.25 and beta=3.15, GUI tool implementation by Ryberg et al., 2021a). To avoid near-source effects we limited our dispersion curve analysis to traces with offsets >1 km and further decimated the data set to ~5000 station pairs, by requiring a minimum spatial receiver station separation. We derived dispersion curves at 30 frequencies ranging from 1.2 to 20 Hz in steps equally spaced with log(frequency). We started picking for every trace when an energetic arrival was present at frequencies between 2 and 5 Hz, from there extending the picking to lower and higher frequencies until a coherent (continuous) dispersion curve branch disappeared. We used the instantaneous frequency to avoid systematic dispersion curve shifts (caused by non-uniform frequency distribution of energy in the trace) for the following inversion step. Fig. 5 shows two typical examples of a dispersion curve for an individual station pair. Despite the fact that a number of traces suffer from poor signal-to-noise ratio, since all stations had been deployed in a densely wooded region, ~45000 travel time picks (frequency dependent travel times of group velocity arrivals)

could be determined. This subset represents ~7% of the dispersion curves of the entire data set (Fig. 4A, 400x400 stations /2 = 80000 station pairs). Bootstrapping inversion tests using an even further decimated data set resulted in very similar final 3D models, so our 7% subset appears to be quite representative. Finally, we used this travel time (or dispersion curve) data set to invert for a three-dimensional shear-wave velocity model below the seismic array.

Instead of using a classical linearized inversion with a Tikhonov-type regularization (including damping, smoothing, a starting model, etc.) we applied a tomographic inversion technique based on Bayesian statistical method (transdimensional, hierarchical Monte Carlo search with Markov Chains using a Metropolis/Hastings sampler, Bodin et al., 2012) to derive the 3-D distribution of shear wave velocity (and its uncertainty), liberating us from the potentially subjective choice of starting velocity models, damping and smoothing parameters, etc. which are necessary when using classical inversion techniques. We follow the fully 3-D technique by Zhang et al. (2018) to invert for a 3D velocity model, but replace the involved travel time calculations by a fast finite-difference based eikonal solver (Podvin & Lecomte, 1991). The latter one, given its high computational speed, is essential when using the Markov Chain Monte Carlo inversion approach since the forward problem (i.e., calculating the travel times for a given model) has to be solved orders of magnitude more often compared to classical inversions.

The 3-D seismic velocity field is discretized by a set of polyhedral Voronoi cells, with a shear wave velocity value assigned to every cell. The model misfit, i.e., the differences between the travel time observed and predicted by the model, is calculated with the following steps: the 3-D model is horizontally gridded (50 m spacing), at every location a vertical 1-D shear velocity model is extracted from the 3D velocity model, then the group velocity dispersion curve is calculated (Herrman & Ammon, 2004; Herrmann, 2013) for the 1-D model, assuming Vp/Vs ratios and density values according to Brocher (2005). When repeated for all locations of the horizontal grid, 2-D velocity maps for all frequencies are constructed. For all 2-D velocity maps, travel times (with respect to the virtual sources and receivers) are calculated using a fast eikonal solver (Podvin & Lecomte, 1991) and eventually the misfit is determined.

The Markov chains are started with a model consisting of a random number of Voronoi cells with random cell centers and shear wave values. By randomly changing the model (changing the velocity and/or position of a cell, adding or removing a cell, etc., details in Zhang et al., 2018) we construct a Markov chain of consecutive models. After a burn-in phase, the (well-fitting) models are decimated (i.e., only every 200[th] model is selected from all well-fitting 15,000 models) gathered for further derivation of a reference solution. For even more extensive model space exploration, models from 1000 separately evolving chains are investigated, easily achieved on computer clusters. The final results, after the burn-in phase (several thousands of well-fitting models) are combined to analyze their statistical properties, i.e., averages and uncertainty.

The inversion is trans-dimensional because the number of Voronoi cells is not fixed, and it is hierarchical because we invert for data noise, i.e., all remaining data misfit which can not be explained by the model. This approach has the advantage that the derived models (distribution of velocities and their uncertainties) are almost completely data-driven, i.e., no prior choice of damping, smoothing, starting model, etc. is needed.

Figure 6 shows the evolution of 1000 Markov chains, both for misfit (B) and model complexity (A, number of Voronoi cells). The advantage of using a hierarchical version of the Markov chain Monte Carlo method resulted in the derivation of the (remaining) data noise for every signal frequency, i.e., the part of the data which could not be explained by the models. This noise is caused by mispicks of the dispersion curves, anisotropy (not considered in the inversion), noisy Green's functions, etc. This noise is typically increased at the upper and lower parts of the frequencies with lower number of data points, thus potentially with poor signal-to-noise ratios, as can be seen in Fig. 7.

## 2.3 Electromagnetic data

For geological interpretation, the passive seismic data are supplemented with resistivity data based on an airborne electromagnetic survey using the Geotech Versatile Time Domain (VTEM™ ET) system. The VTEM™ ET transmitter-receiver loop was in concentric-coplanar and Z-direction oriented configuration and was towed at a mean distance of 30 meters below a helicopter. The survey was flown in a south to north, 40° NW direction, with a traverse line spacing of 50 meters.

Fifty-four time measurement gates were used for the final data processing in the range from 5.79 to 9481 μsec. The Full Waveform EM specific data processing operations included half cycle stacking (performed at time of acquisition), system response correction as well as parasitic and drift removal. In addition, a three-stage digital filtering process was applied to reject major sferic events and the signal to noise ratio was further improved by the application of a low-pass linear digital filter.

The VTEM ET data were reprocessed to eliminate coupling with mapped infrastructures, to increase S/N, and assign noise. The data were inverted with spatial constraints in a layered earth approach (SCI). Where present, Induced Polarization effects were taken into account, using the dispersive resistivity model of Cole and Cole (cf. Viezzoli et al., 2017). The result is a 3D distribution of Cole Cole parameters, the most relevant being direct current resistivity and chargeability.

## 2.4 Data Integration

In order to facilitate a comparison between the VTEM resistivity data and the seismic velocities, the inverted EM data, represented by a layer thickness with a corresponding resistivity value, was discretized along each 1D resistivity-depth array to a resolution of 2 meters and cropped to the depth of investigation. Subsequently, Bentley Systems Leapfrog Geo (version 5.1.1) was used to create a numerical model for the VTEM point data at 10 m resolution using spheroidal interpolation with spatial constraints including a 25 m distance buffer and topography. Similarly, the seismic velocity data with an original point spacing of 10 m was interpolated using a linear interpolant using the same spatial constraints as the VTEM data. Finally, both parameters were mapped into one and the same block model with 15x15x15 m cells.

To systematically evaluate the spatial distribution and geological significance of specific value ranges within this integrated block model, an unsupervised classification was performed on the bivariate 3D dataset (Fig. 8). The vector of variables at

each grid cell was extended with its neighborhood to account for the spatial structure of the data (spatial clustering cf. Talebi et al., 2020). Two different neighborhoods were considered: one containing the cross of 6 grid cells immediately in contact with the target grid cell, and one containing the 26 cells of a cube surrounding the target. Additionally, both spectral clustering (Ng et al., 2002; von Luxburg, 2007) and conventional k-means were applied to both neighborhood settings. Finally, a 9 groups classification based on k-means with 26 immediate neighbors was chosen because it provided a well-balanced distribution of data into clusters with the largest spatial coherence. K-means was chosen as a segmentation algorithm rather than a density-based clustering approach to not exclude value ranges that are only represented by a few points.

## 3. Results and discussion

### 3.1 Ambient noise inversion results

Instead of running a single Markov chain, we simultaneously investigated 1000 chains in parallel. All of them had a different, randomly chosen starting model. The evolution of the models along every chain (Fig. 6) shows that the data misfit quickly decreases, and reaches stable values after ~3000 models, while the model complexity (= number of Voronoi cells to describe the individual model) still increases. The latter starts to stabilize from model number ~5000. From model number 10000 we assumed that the burn-in phase was finished, and stable systematic model sampling in the model space was reached. All models beyond 10,000 have been collected and analysed by calculating average S-wave velocities and their uncertainty at every location in the 3D model. The velocity model is generally characterized by strong velocity anomalies (Fig. 9A,C). It is a rather smooth model with gradual velocity variations as it is typical for tomographic inversions (i.e., not showing first-order discontinuities). The velocity uncertainty (Fig. 9B), as expected, is quite low in shallow depth ranges, and quickly reaches larger values (red colors) at greater depth, a typical behavior when inverting surface wave data. This uncertainty (velocity resolution) has to be taken into account when interpreting the recovered S-wave velocities.

The shear wave velocity distribution shows strong lateral contrasts and is well resolved down to approx. 500 m below the surface. The following velocity anomalies can be delineated:

- SA1: The seismic velocity model from the surface down to a depth of ca. 400 m is marked by relatively high shear velocities in the NW portion, and lower shear velocities in the SE portion of the area, giving rise to a prominent NW-SE spatial trend. The differences between these regions are expressed most strongly at the northern end of the study area, where adjacent high and low-velocity zones form a pronounced SW-NE striking, vertical discontinuity (anomaly SA1, Fig. 9A2). This discontinuity aligns well with the dominant tectonic grain of the area and the SW-NE striking, steep geological contact between two-mica schist hornfels within the contact metamorphic aureole of the granite (high velocities) in the NW and mica schist / muscovite gneiss (lower velocities) due SE (see Fig. 1).

- SA2: In the southern part of the study area, the prominent NE-SW spatial trend described above (SA1) is crosscut (and offset?) by a NW-striking linear zone, extending from surface to about 100 m depth (anomaly SA2, Fig. 9A1,

C1). This zone coincides with a known, regionally important fault zone (the Greifenbach fault, see Fig. 1A). It also marks the trend of the Greifenbach stream and associated Quaternary deposits.

- SA3: Two lobe-shaped, NW-dipping low-velocity zones occur in the two-mica schist at depths down to ca. 400 m (anomaly SA3, Fig. 9A3, C2). These lobes are orientated parallel to the main metamorphic fabric and are also imaged by resistivity data (anomaly EA2).

Interestingly, the seismic data only broadly images the granite-mica schist interface, and we observe both high- and low-velocity zones cutting obliquely across this lithological boundary. Thus, the observed velocity distribution is likely related to factors that are superimposed on the primary lithology, such as weathering, alteration and/or mineralization.

## 3.2 Electromagnetic data

The VTEM airborne electromagnetic partially overlaps with the area of the passive seismic survey (Fig. 10A), and provides valuable complementary information. In depth, the resistivity model is constrained by the depth of investigation, which varies from ca. 50 to 600 meters, depending on geology, EM system attitude, noise in the data, and inversion stability. Within the study area, i.e., the bounds of the seismic array, we observe the following resistivity anomalies (Fig. 10):

- EA1: A conductive top layer that shows a good correlation with the mapped extent of Quaternary sediments and cover thickness extracted from drill cores (stippled lines in Fig. 10B–D).
- EA2: NNE-SSW striking high-conductivity zones in the eastern part of the study area, corresponding to the location of muscovite gneiss intercalations in the two-mica schists. The 20–30° northwesterly dip of these anomalies follow the approximate trend of the main foliation (Fig. 1). EA2 is also imaged by the passive seismic data (anomaly SA3).
- EA3: Four to five distinct high-conductivity zones, circular in horizontal section (Fig. 10A), and plume-shaped in cross-section (Fig. 10C). These anomalies are related to anthropogenic features, i.e., game fences to protect trees in forest cultures, confirmed by field validation. The presence or absence of an anomaly at the fence is due to the fence characteristics, e.g., density and quality of the fence stranding (metal weaving), whether grounded to the earth, orientation, size, metal quality, etc.

## 3.3 Integration and geological significance

The identified anomalies in velocity and resistivity space provide valuable insights into the subsurface structure of the study area. To investigate anomalies in the bivariate space and facilitate geological interpretation, particularly in terms of mineralization, a subdivision of value ranges of the combined 3D dataset using a spatially-constrained K-means clustering approach is applied. The resulting 9 classes are spatially relatively coherent (Fig. 11C, 12C) and integrate both the shear velocity and resistivity structure. For example, the classification clearly reproduces the gently NW-dipping low-resistivity zones (Fig. 12, rows 1 and 4) as well as the ±vertical contacts between zones of high and low shear velocities (Fig. 12, rows 1, 3, 6 and 7). The classes exhibit distinct ranges of shear velocities and resistivities (Fig. 8B, Fig. 13A), some of which can be associated with identified anomalies in velocity and/or resistivity space. Cluster 1 has some of the highest resistivities and

seems to be spatially associated with two-mica schists on the SW side of the Greifenbach fault. Cluster 2 has the lowest values of both resistivity and shear velocity and corresponds to the seismic anomaly SA3 and resistivity anomaly EA2. It exhibits a prominent NW-dip and together with the less conductive cluster 6 most likely corresponds to the muscovite gneiss intercalations in the two-mica schists. Cluster 3 corresponds with the seismic anomaly SA2 and resistivity anomaly EA1. It is mostly confined to the surface and thickening along the Greifenbach fault, and is therefore interpreted to be associated with faulting-related fracturing and Quaternary deposits. Cluster 4 exhibits the highest shear velocities of all clusters and occurs mostly in the apical part of the granite on the NE side of the Greifenbach fault. Cluster 5 spatially coincides with the mapped two-mica schist within the contact metamorphic aureole of the granite. Cluster 7 exhibits intermediate resistivity and shear velocity values, and, occurring mostly in the endo-contact part of the granite, may represent the normal, non-mineralized granite together with cluster 4. Cluster 8 is characterized by intermediate shear velocity and high resistivities, and is located at the granite interface, where it spatially coincides with most of the drilled greisen occurrences in the considered volume (Fig. 11C, 12C, 13). Cluster 9 is characterized by relatively low velocities and resistivities and spatially coincides with the anthropogenic plume-like conductive EM anomalies EA3 (Fig. 10C, 11B, 12B).

A number of clusters (1, 3, 4, 5 and 8) coincide with drilled occurrences of greisenized rocks (Fig. 13B). The cluster spatially most closely associated with the drilled greisen occurrences is cluster 8, which has a width of up to 750 m, a length of 1350 m and a thickness of 200 m (Fig. 11, 12). Its lateral dimension is constrained to a certain degree by the model boundaries, the chosen classification parameters and the inherent smoothness of the 3D inversion models. The mean shear velocities and resistivities of this cluster (avg. = 2.4 km/s, 3.8 Ωm) are relatively high compared to those of the classes representing the surrounding non-greisen rocks (classes 2, 4, 6 and 7, avg. = 2.1 km/s, 2.9 Ωm). This is in line with the petrophysical constraints outlined above. Although speculative given the described uncertainties, our data thus hint at the presence of a large greisenized zone at the periphery of the Geyer granite in depths of 50–250 m underneath the surface. Such a body is currently not documented, but greisen bodies of similar dimensions as those of cluster 8 are known from the Ehrenfriedersdorf area at the Sauberg and Vierung prospects (Brosig et al, 2020). In light of the fact that the Erzgebirge is considered underexplored with regard to its Li mineral resources (Dittrich et al., 2020), the stage-1 greisen zones in the study area, which are considered to be barren with respect to Sn-W (Hösel et al., 1994; Jung and Seifert, 1996), may simply not have received much attention.

The geophysical exploration of greisen ores in the Geyer-Ehrenfriedersdorf area is a challenge because of their occurrence in a populated area, their petrophysical indistinctiveness and limited size. As demonstrated in this study, some of these challenges can be overcome by using a combination of different low-impact geophysical methods complemented by geological and petrophysical data. Our analysis demonstrates the potential of 3D passive seismic data from dense networks for the geological characterization of the subsurface in a crystalline rock environment. Passive seismic data is particularly well suited for application in populated areas, because of its low environmental impact and the fact that it is unperturbed by anthropogenic electromagnetic noise, which is a problem for magnetic and electromagnetic prospecting (e.g., Morris et al., 2021). Nonetheless, as shown in this study, airborne time-domain electromagnetic data, which was used to identify the

cover-basement interface and conductive layers in the metamorphic host rocks, can provide useful indirect constraints as complementary information for the delineation of greisen mineralization.

**4. Conclusions and outlook**

We conducted an ambient noise seismic study in the Geyer area to gain insight into the 3D structure of the subsurface and evaluate the region's potential for greisen-hosted, polymetallic mineralization. Based on a novel 3D inversion technique that has been applied to a passive seismic dataset, supplemented by and integrated with 3D resistivity data from airborne time-domain electromagnetics we are able to:

1.   identify distinct shear velocity anomalies down to a depth of ca. 500 m that most likely correspond to primary
geological features such as the transition from mica-schist hornfels to muscovite gneiss dominated host rocks, and near-surface fracturing related to strike-slip faulting;

2.   use resistivity anomalies to delineate the cover-basement interface as well as conductive zones parallel to the main fabric of the metamorphic host rocks;

3.   employ spatial clustering of the combined seismic-EM dataset and drill core validation to identify parameter ranges
corresponding to a potential greisen mineralization.

The study showcases the benefits of a combined use of low-impact, geophysical data collection and processing technologies for mineral resources exploration. Crystalline settings (as in the Geyer-Ehrenfriedersdorf case) are particularly challenging for high-resolution controlled-source seismic. The resolution of the ambient noise velocity models is lower than those expected for reflection seismic imaging in a sedimentary setting but passive surveys can provide very valuable constraints at
reasonable costs for an enhanced understanding of subsurface architecture and the distribution of ore bodies, particularly if integrated with other geophysical parameters.

The study area turned out to be a well-suited test area because of the large amount of information on the subsurface due to the existence of numerous archive data that allowed the benchmarking of the results and a geologically consistent interpretation.

The passive seismic data set was gathered with a large network of regularly spaced seismic sensors but only a small part of the entire data set was used to derive a 3D velocity model. By using other subsets of the entire data set, optimized experiment designs, i.e. number and geometric distribution of seismic sensors could be developed and evaluated for future studies.

**Data availability**

Seismological data is archived at the GEOFON/EIDA archive (Haberland et al., 2021). The velocity-resistivity 3D block model is available under Ryberg et al. (2021b) and viewable under https://bit.ly/3B2aEvD.

**Author contribution**

TR designed the field methodology of passive seismic investigations, developed the inversion software, performed the passive seismic inversions, participated in the field work, and contributed to the original manuscript preparation. MK contributed to concept development, participated in field work, conducted the integrative, geological analysis and substantially contributed to writing the manuscript. CH contributed to the inversion methodology, participated in the field work, data curation and contributed to the original manuscript preparation. RT coded and performed the spatial clustering analysis and contributed to writing the manuscript. AV performed QC, processing, and inversion of the electromagnetic data and contributed to the preparation of the manuscript. RG assumed management and coordination responsibilities for this project, helped to design the experiments, acquired part of the funding that led to this publication, and contributed with editing of the manuscript. We would like to thank N. T. Arndt and the anonymous reviewer for providing comments that helped to improve the manuscript.

**Competing interests**

The authors declare that they have no conflict of interest.

**Acknowledgements**

Instruments for the seismic network were provided by the Geophysical Instrument Pool Potsdam (GIPP, GFZ), grant GIPP202010. The inversion of seismic data was carried out on the high-performance computing cluster at GFZ. Funding was provided by the GFZ and by the European Union's Horizon 2020 research and innovation programme under grant agreement No 776487. We gratefully acknowledge Geotech Ltd. for the acquisition and pre-processing of the EM data. Sam Thiele and Sean Walker are gratefully acknowledged for their help with data processing. Thanks also to Leïla Ajjabou, Yuleika Madriz, Erik Herrmann, Hernan Flores, Thomas Lüttke, Juan-Felipe Bustos, Ariane Siebert, Falk Brethauer, Matthias Bosdorf and Michael Weber for their help in the passive seismic field campaign. MK acknowledges Tobias Duteloff (Sächsisches Landesamt für Umwelt, Landwirtschaft und Geologie) for his kind assistance and provision of archive material. Furthermore, we thank the city of Ehrenfriedersdorf for their support and hospitality during field work.

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

**Figures**

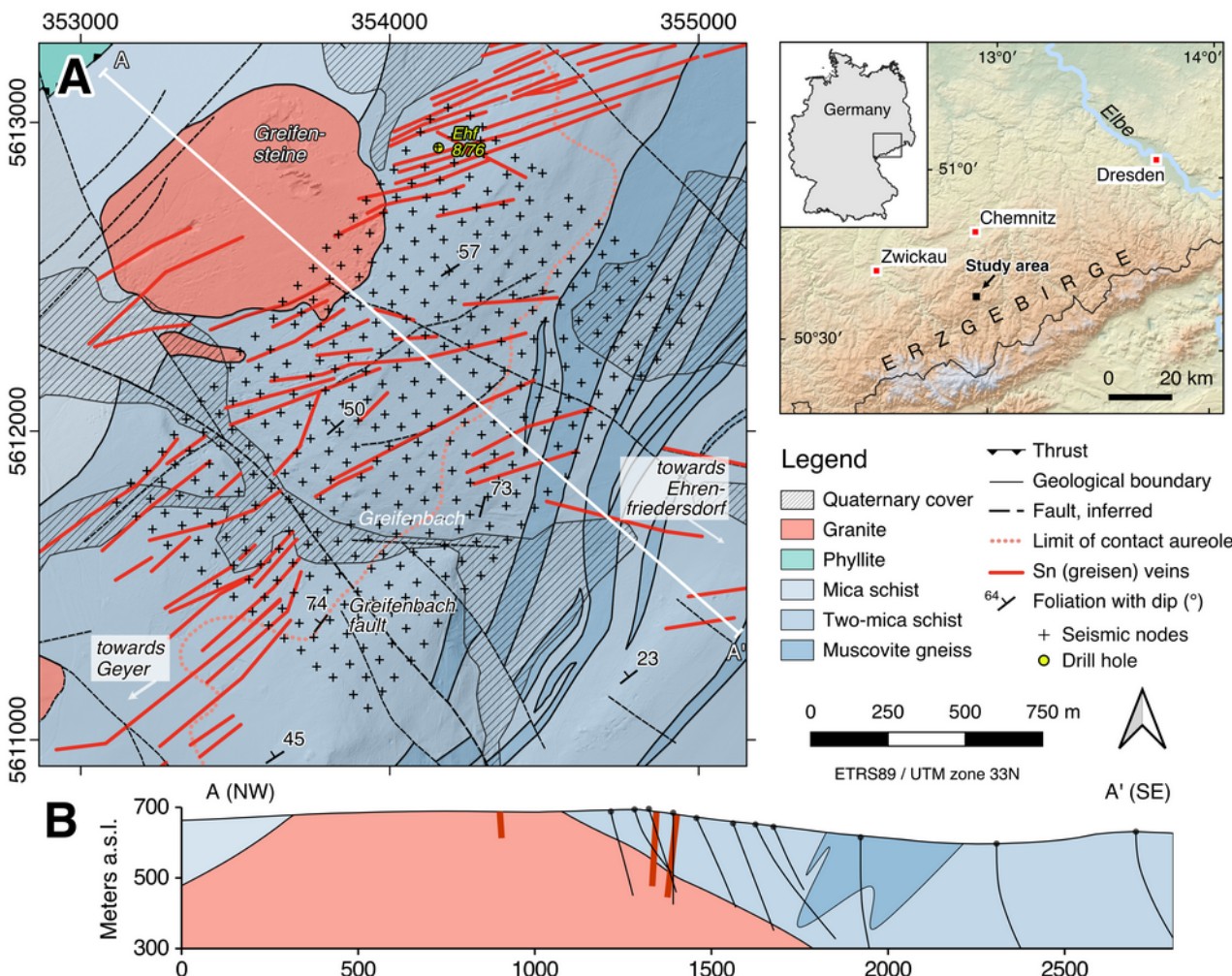

**Figure 1: (A) Geologic map and (B) cross-section of the study area. Sources: Sächsisches Landesamt für Umwelt, Landwirtschaft und Geologie (Geological Survey of Saxony); Kirsch and Steffen (2017). Drill holes in the section were projected from a distance of 150 m. Crosses in A indicate the location of the seismic stations.**

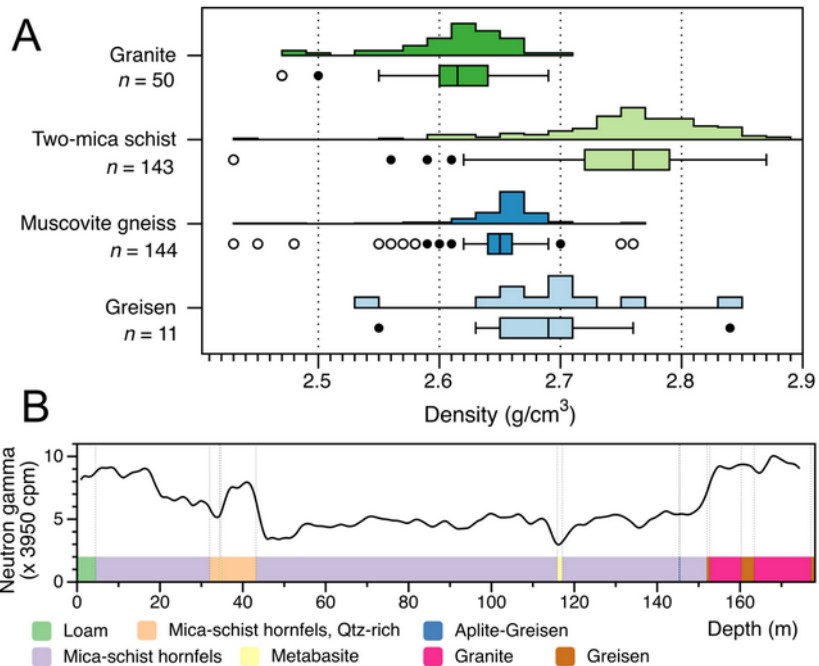

Figure 2: Petrophysical data for the rock types of this study. (A) Density based on drill-core samples from the Ehrenfriedersdorf-Geyer area (data source: Klemps and Lindner, 1985); (B) Downhole neutron gamma log data from drill hole Ehf 8/76 (see Fig. 1 for location). Data source: Sächsisches Landesamt für Umwelt, Landwirtschaft und Geologie.

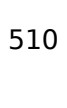

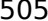

Figure 3: Frequency and directional beamforming analysis (exemplary). a: Spectrogram of data of station 073 at day 2020-07-31 (vertical component). Power spectral density is color coded. b and c: Beamforming analysis within two time windows of 1 hour length between 5:00 and 6:00 UTC and 19:00 and 20:00 UTC (indicated in panel a) using a subset of 22 stations (vertical components). Normalized beampower is color coded. Radius is slowness in s/km (from 0 s/km in the center to 0.5 s/km at the edge), azimuth is direction from north. For each time window beamforming analysis has been performed in three frequency ranges 1.2 - 4.5 Hz, 4.5 - 10 Hz and 10 - 20 Hz. See text for details.

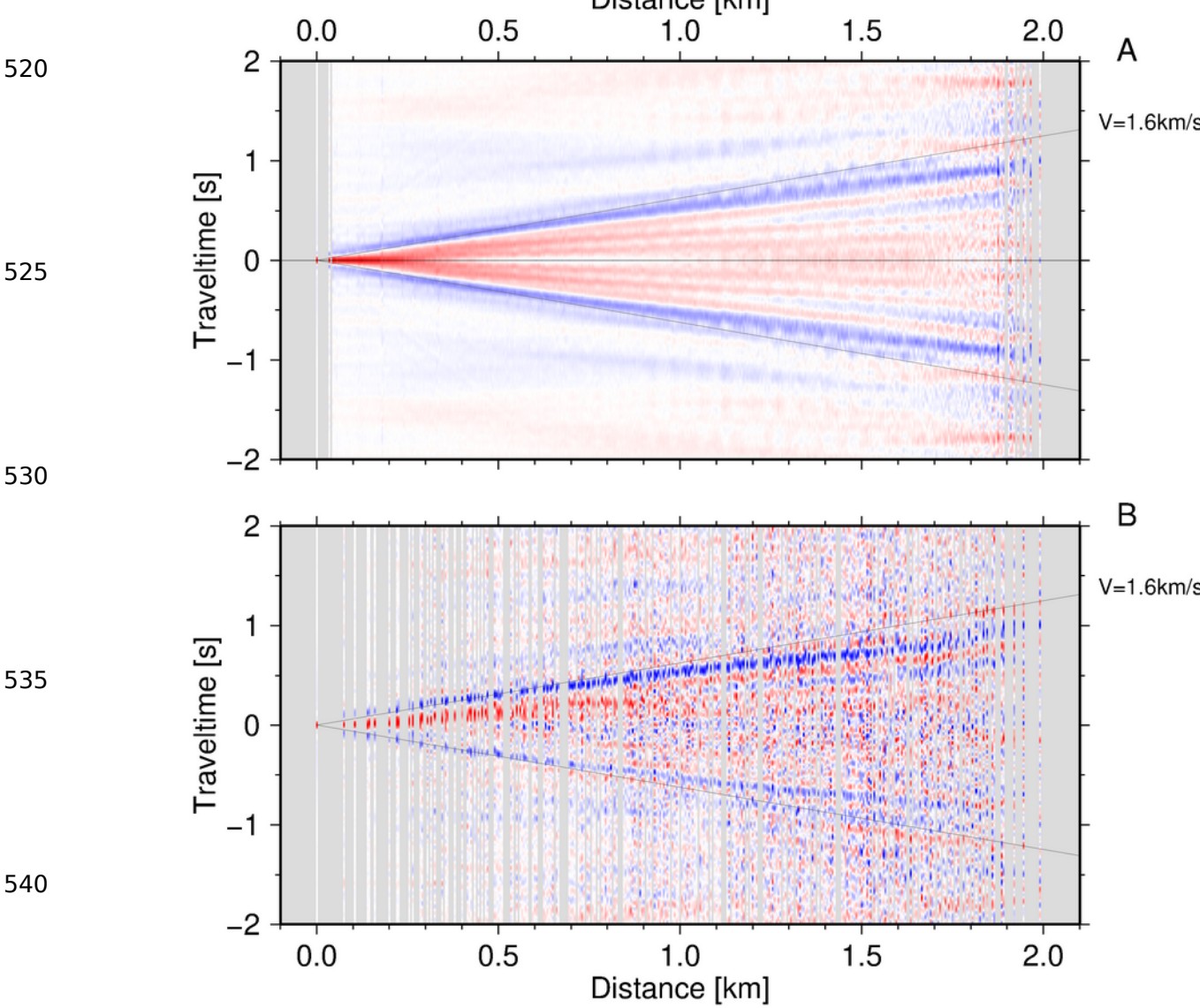

**Figure 4: Cross-correlations of the vertical component data of all stations (A) against each other sorted by distance in order to evaluate the data quality. The ambient noise technique allows to recover the Green's function for station pairs from the recorded ambient noise. Shown is a stack of all unfiltered Green's functions, stacked in offset (distance between stations) bins. Surface waves are clearly identified and dominate this record section (red/blue bands starting at zero distance). B shows the virtual shot gather between the westernmost station (001) against all other stations. The strong asymmetry between the causal and acausal phases is caused by the non-uniform distribution of noise sources.**

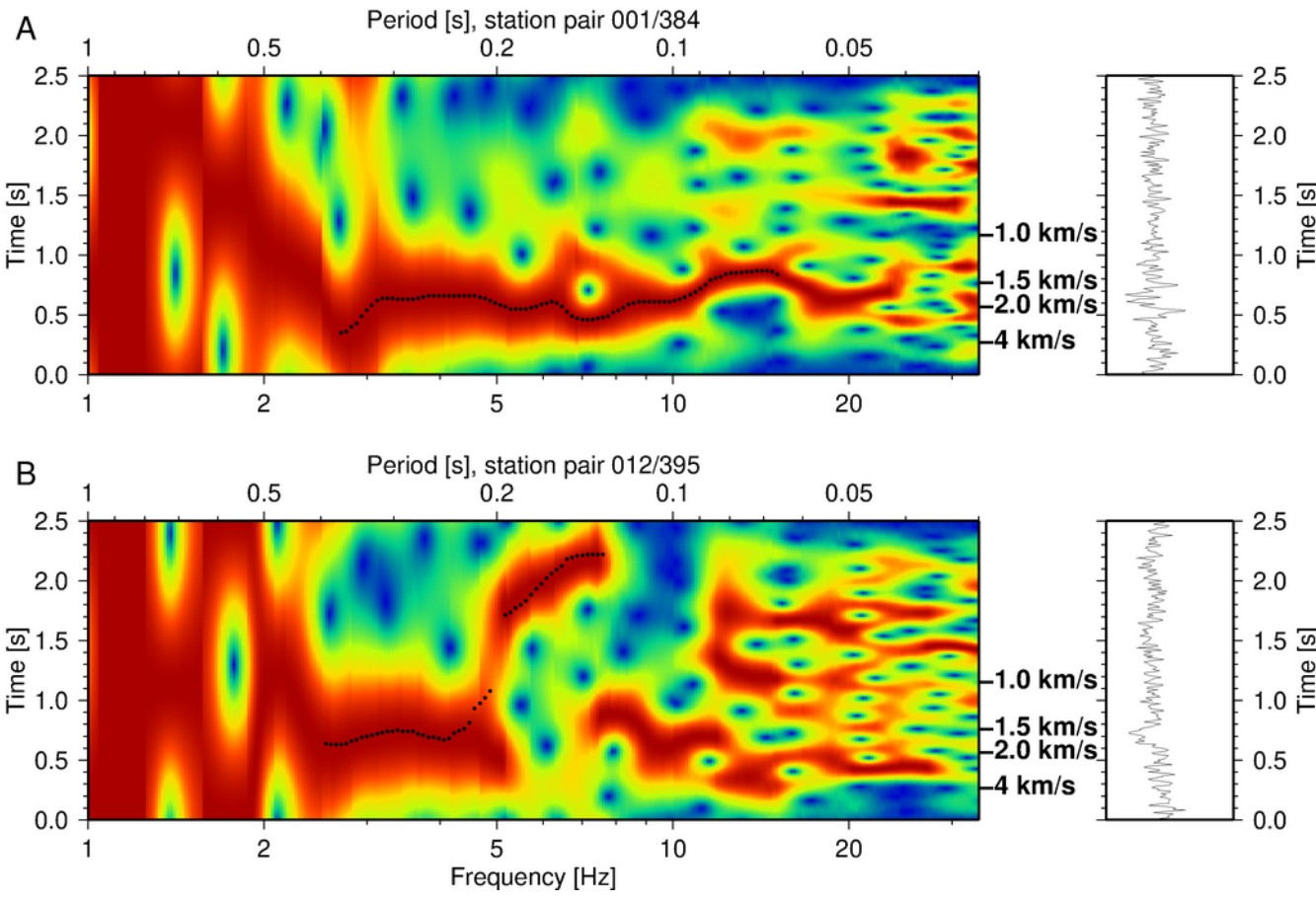

Figure 5: Example of dispersion curve analysis for two traces (left panels) of the cross correlation of the vertical component seismic records (right panels) between station 001 and 384 (A) and station 012 and 395 (B). The dispersion curve analysis is in the left panel (red colors indicate high, blue colors indicate low amplitudes). Traces are normalized in every frequency/period band, travel time picks (i.e. the dispersion curve) are indicated by black dots. Corresponding group velocities are indicated at the right side of the panel. A dispersed surface wave curve (Rayleigh phase) can be identified for signal periods between 3 and 15 Hz (A) and ~2.5 and 7 Hz (B). Station pair 012/395 crosses a region of low shear wave velocity.

555

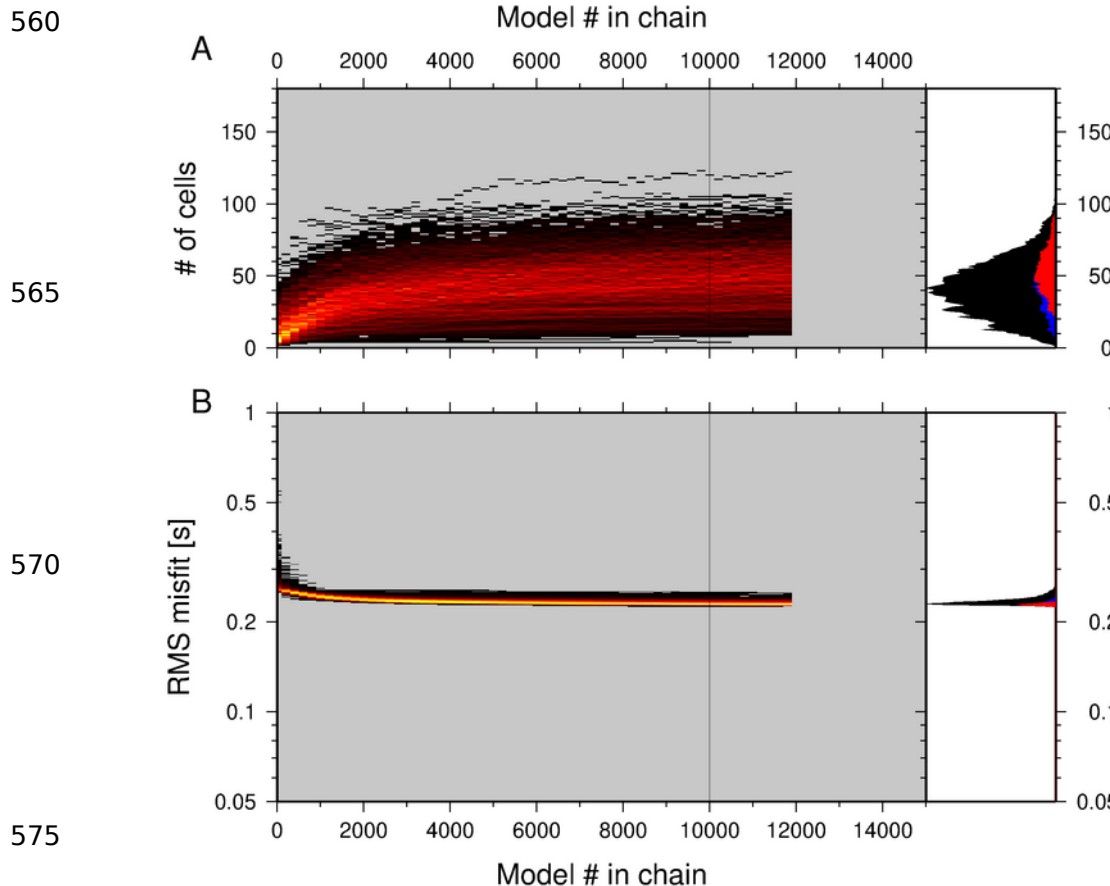

**Figure 6: Evolution of misfit along Markov chains. Shown is the histogram distribution of the data misfit (B) and model dimension (number of cells, A) during the evolution along 1000 Markov chains using a heat map color scale. The black line at 10000 shows the end of the burn-in phase, following models are used to derive the final model. Relative histogram plots of the distribution of data misfit (bottom) and model dimension (top) are added at the right side, with all models shown in black, blue models are the post-burn-in ones, red are the 90% best fitting models. The best fitting models (red) are typically characterized by higher model dimensionality. The forward problem (travel time calculations) was solved for more than 3.5 x 10$^8$ models. Note the log scale for the iteration number and data misfit.**

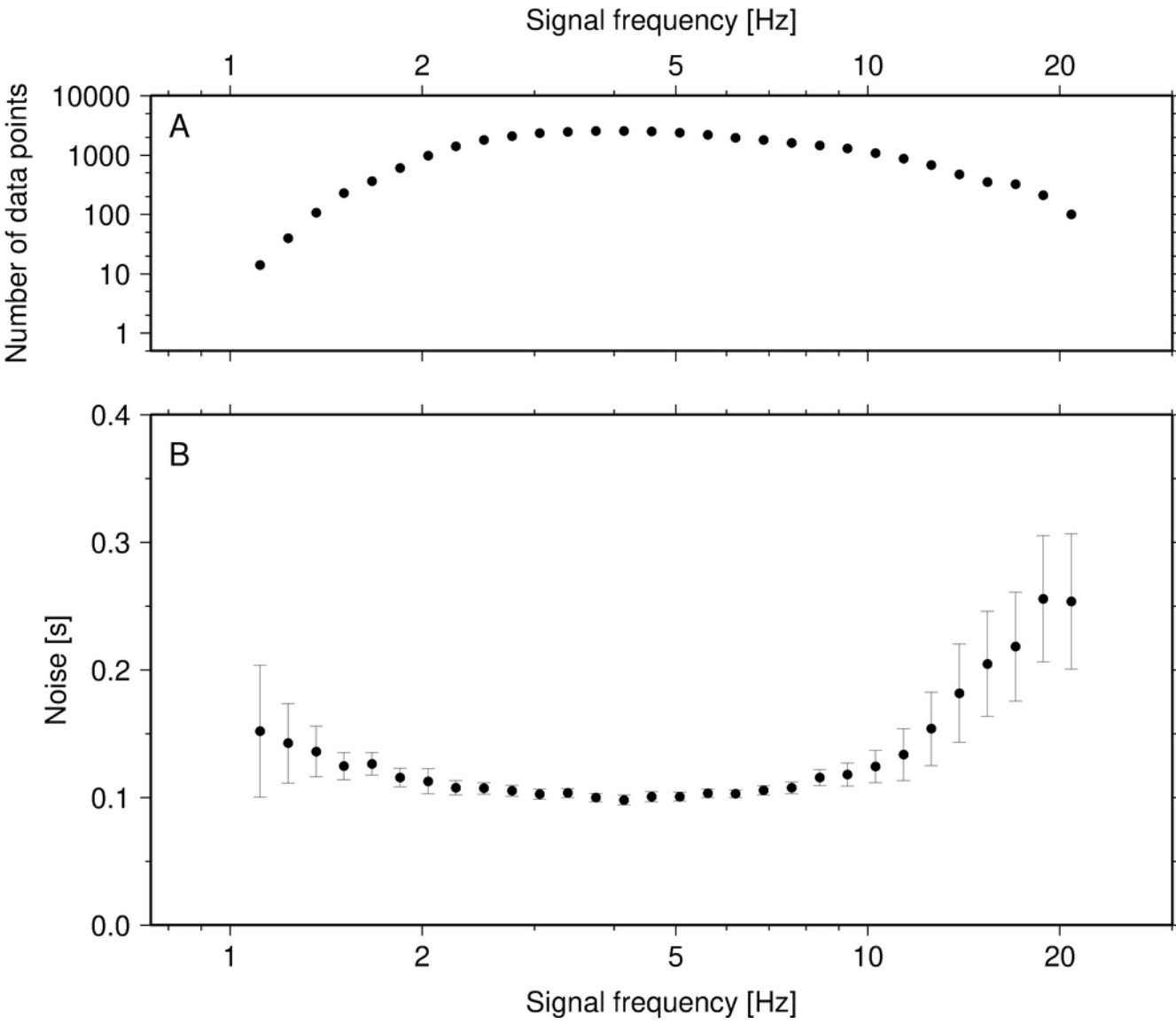

**Figure 7. The hierarchical McMC method allows for inversion of data noise. We inverted for the frequency dependent data noise: Bottom (B) is shown the recovered data noise at every signal frequency analysed. Top (A): is shown the number of data points (travel times/group velocities). In the frequency range between 2 and 10 Hz, corresponding to at least several hundred data points per frequency, the noise level is small: i.e. this part of the data contributes significantly to the derivation of the final model.**

590

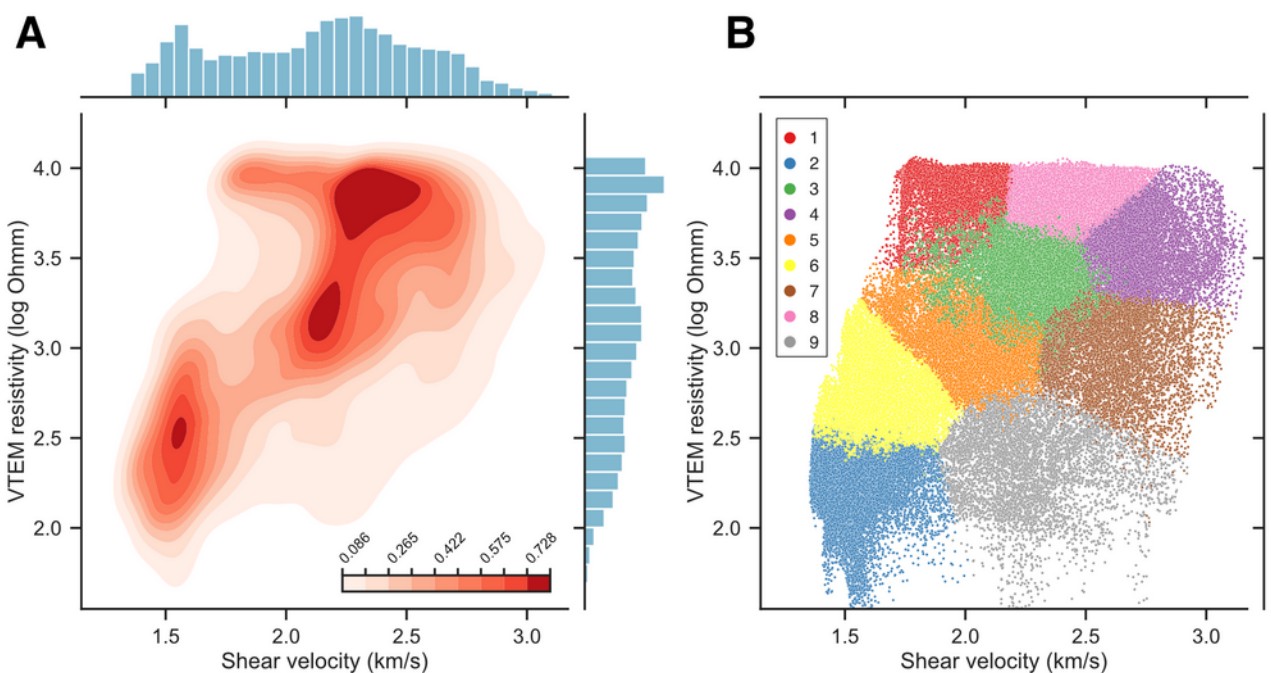

**Figure 8: (A)** Bivariate kernel density estimate (KDE) contour plot and marginal histograms of seismic velocity vs. resistivity. **(B)** Scatterplot of seismic velocity vs. resistivity showing class labels of classification.

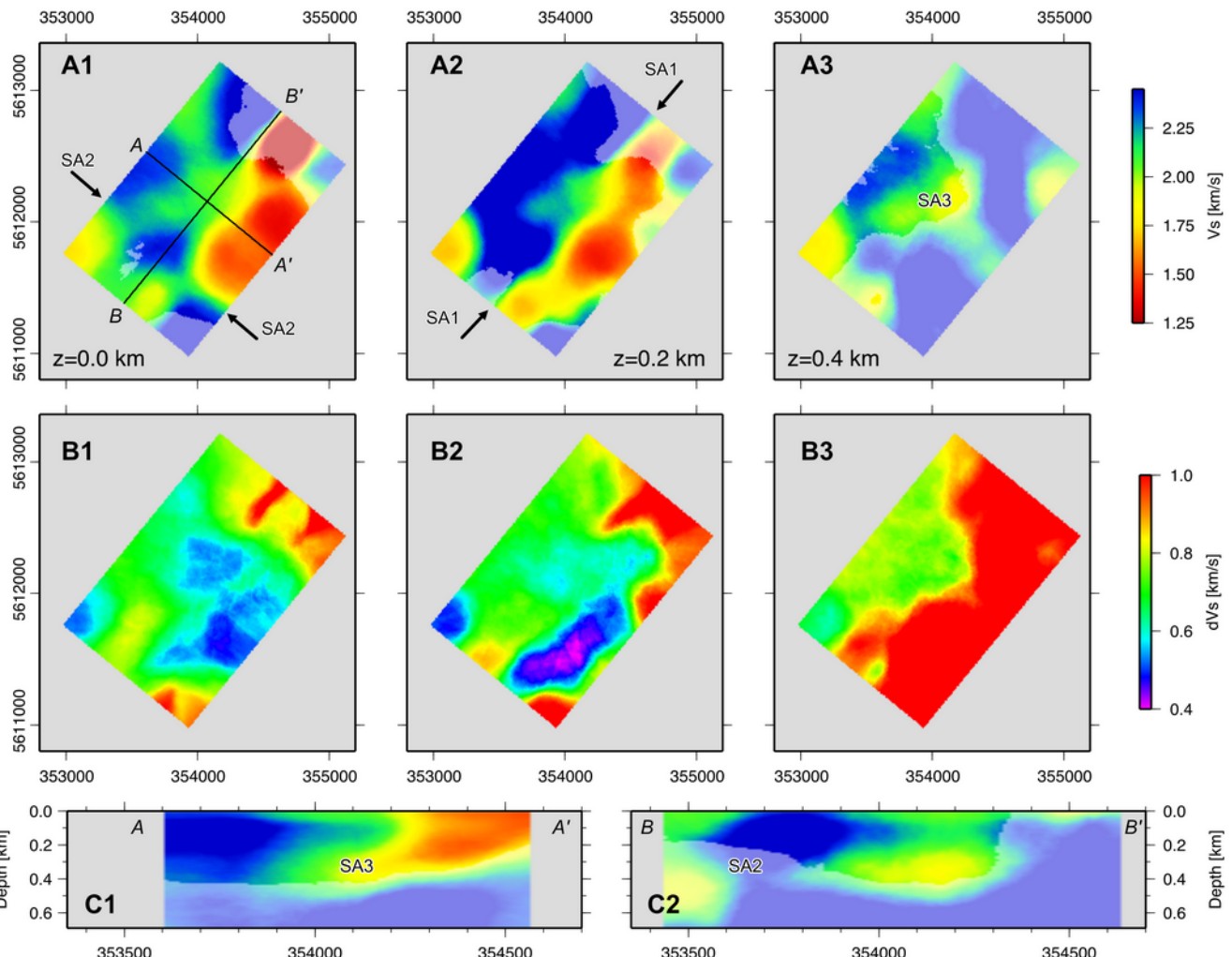

Figure 9. Slices through the final three-dimensional shear velocity model. (A) S-wave velocity, (B) its uncertainty at surface, and at 200 and 400 m depth, respectively. (C) Vertical slices through the velocity model in NW-SE (AA') and SW-NE (BB') direction, respectively. Regions with low color saturation indicate a high velocity uncertainty (above 0.8 km/s). Labels SA1, SA2, and SA3 mark velocity anomalies described in the text.

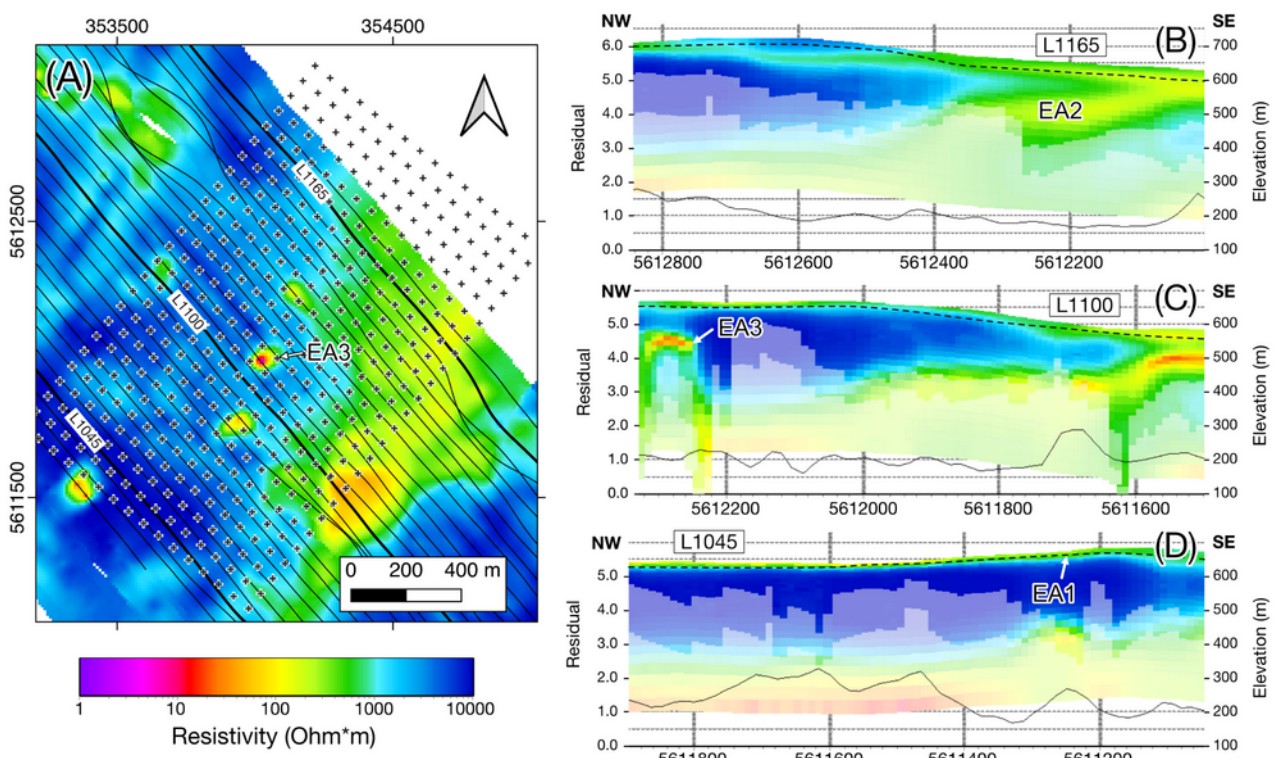

**Figure 10. VTEM™ ET resistivity data. (A) Map of the study area including seismic station array (crosses) and VTEM flight lines (black lines). Also shown is the resistivity distribution at 70–80 m depth (obtained with Kriging using an exponential variogram). (B)–(D) Resistivity sections along selected flight lines (highlighted in A). Sections have been clipped to the extent of the seismic array. Black line represents data misfit (read against left axis). The white shading in the section indicates the estimated depth of investigation (DOI). Stippled line indicates bottom of Quaternary cover as interpolated from drilling (source: Sächsisches Landesamt für Umwelt, Landwirtschaft und Geologie).**

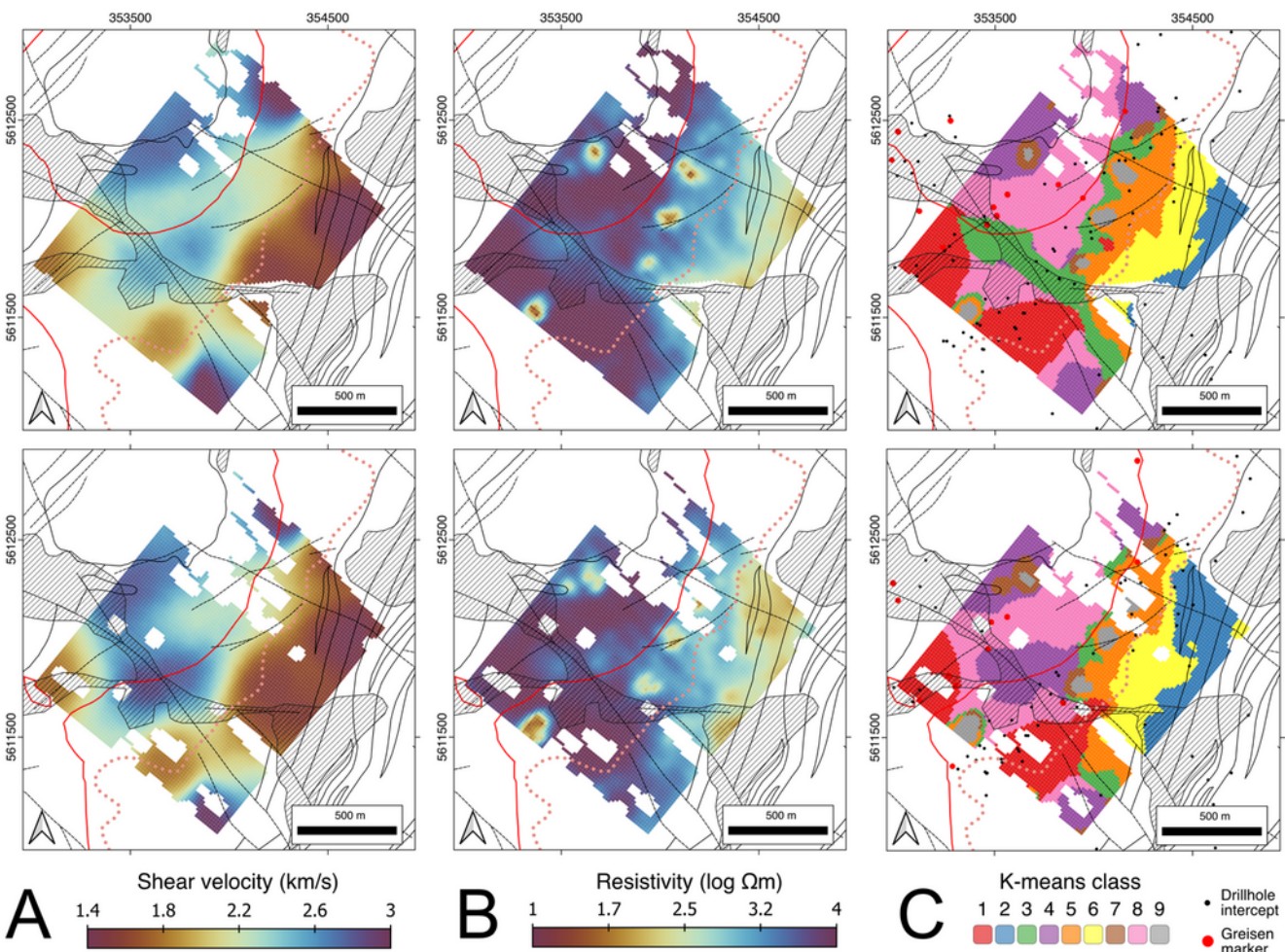

**Figure 11: Horizontal sections through the combined shear velocity–resistivity block model at elevations of 560 m (top) and 510 m (bottom), respectively, showing (A) shear velocity, (B) resistivity, and (C) classification. Overlay shows geological boundaries from geological map (Fig. 1). Red line represents the granite–two-mica schist interface at the corresponding depth as extracted from the 3D geological model (Kirsch and Steffen, 2017).**

610

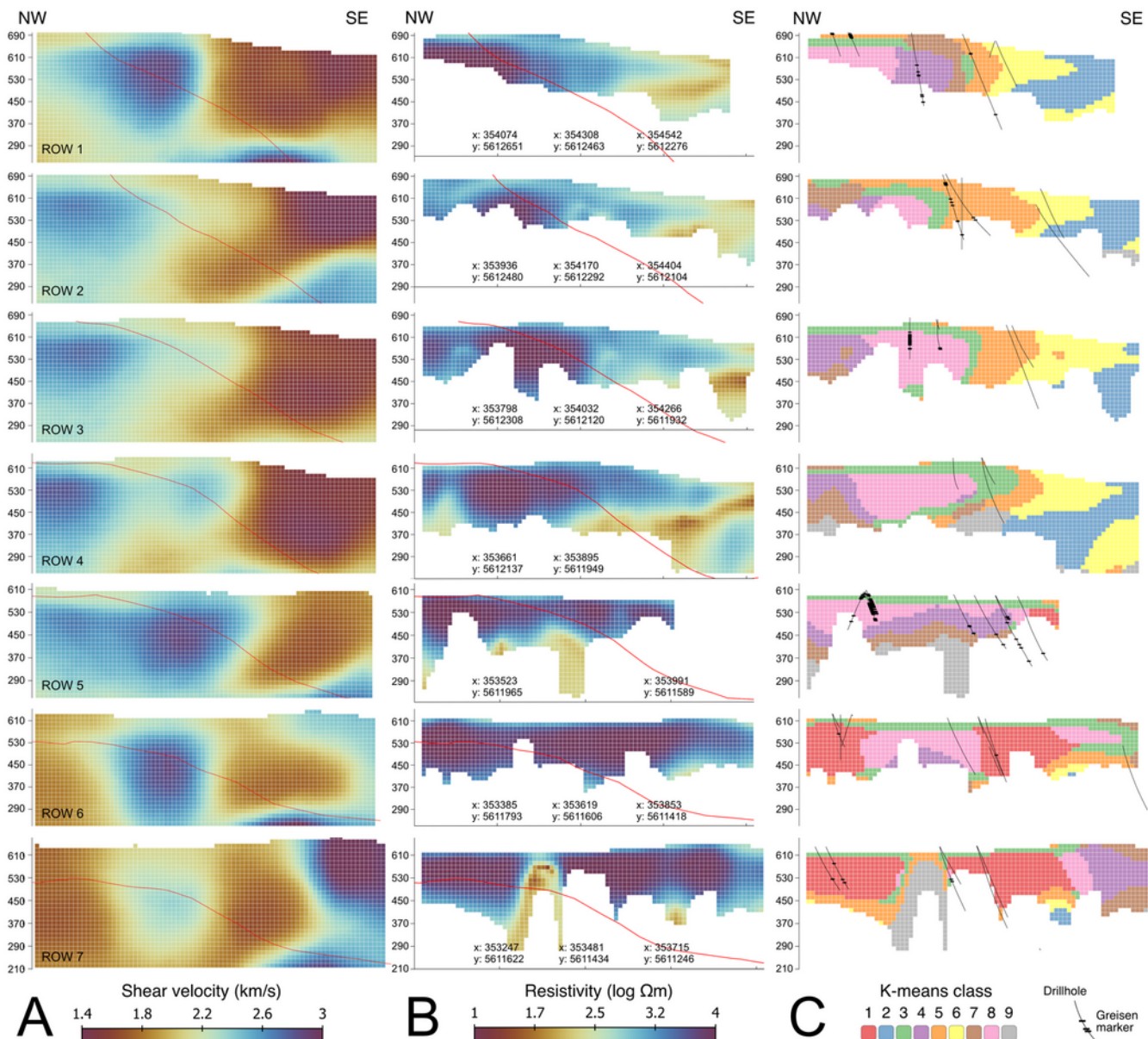

**Figure 12:** Vertical sections (220 m spacing) through the combined shear velocity–resistivity model showing (A) shear velocity, (B) resistivity, and (C) classification. Sections are oriented NW-SE. Northernmost section is at the top, southernmost at the bottom. Red line marks the transition between granite (NW) and two-mica schist (SE). Drill holes and greisen markers in (C) are projected into the section from a distance of up to 50 m.

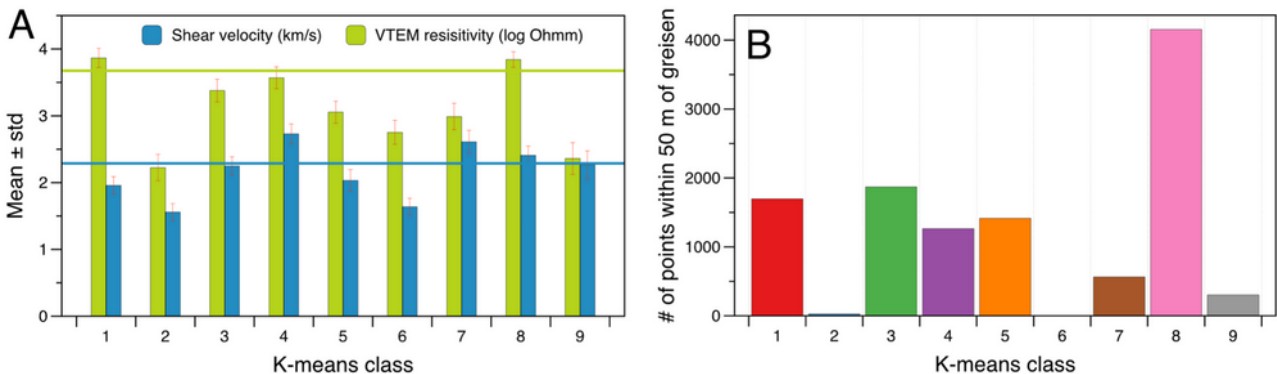

**Figure 13: Classification statistics showing (A) per-class mean values of shear velocity and VTEM resistivity and (B) number of points per class within a 50 m radius of individual greisen occurrences. Horizontal lines in (A) correspond to mean shear velocities and resistivities extracted from block model that are calculated using a buffer of 5, 10, 50 and 100 m around drilled greisen occurrences.**

620