# Peer review of "Ambient seismic noise analysis of LARGE-N data for mineral exploration in the Central Erzgebirge, Germany"

_Solid Earth, 2021_

## Referee Comment (RC2)

SA1

SA2

353000 354000 355000

Greifen-steine

Greifen-steine

*B*

57

50

73

*towards Ehren-friedersdorf*

*Greifenbach*

74

*Greifenbach fault*

*towards Geyer*

45

Germany

13°0'  14°0'

Elbe

Dresden

51°0'

Chemnitz

Zwickau  Study area

50°30'  E R Z G E B I R G E

0        20 km

**Legend**

Thrust

| | |
|---|---|
| Quaternary cover | Geological boundary |
| Granite | Fault, inferred |
| Phyllite | Sn (greisen) veins |
| Mica schist | $^{64}$ Foliation with dip (°) |
| Two mica schist | + Seismic node locations |
| Gneiss intercalations | |

0    250    500    750 m

ETRS89 / UTM zone 33N

A1

353000 354000 355000

A'

SA2

B

B'

A

SA2

z=0.0 km

A2

SA1

SA1

z=0.2 km

B

Meters a.s.l.

A (NW)                                                                 A' (SE)

700

500

300

0        500        1000        1500        2000        2500

[Figure]

A

A (NW)

B

A1

SA2

A'

B

B'

z=0.0 km

A2

SA1

SA1

z=0.2 km

NW                                                    SE

Greifen-steine

Greifenbach

Greifenbach fault

towards Ehren-friedersdorf

towards Geyer

Legend

Meters a.s.l.

Shear velocity (km/s)

1.4    1.8    2.2    2.6    3

[revised manuscript text omitted]

---

## Author Comment (AC1)

References to figure numbers in the reply are related to the originally submitted manuscript. Since figures had been added to the modified manuscript, figure numbers are changed accordingly.

>**General Comments**

>...

>The Empirical Green Functions (EGF) in Figure 2 are very symmetrical. So, it looks like the
>acausal and causal parts are stacked up, right? Or, which part of the EGF was used to calculate
>dispersion curves?

Yes, the Fig 2 IS symmetric, since we show cross correlations of ALL stations against ALL stations! To show that the individual correlations are indeed non-symmetrical, we modified Fig 2 by adding a single virtual shot gather, which clearly shows the asymmetry between causal/acausal phases. We modified Fig. 2.

>A noise directionality analysis (e.g., beamforming) would improve the understanding of the noise
>sources that contribute to the emergence of EGF. 10 days of continuous noise is enough to extract
>the FGE robustly?

We added a new figure showing the frequency and time dependence and the directionality of the noise sources.

We are aware of the fact that the 10 days of noise observation are quite short. Judging exclusively from the appearance of the dispersion curve of a single cross correlation trace, even with relatively short noise observations, dispersion curves (or part of them) can be picked quickly at frequencies dominating the noise. "Adding" more noise records (more observation days) "extends" the frequency range where reliable dispersion curve picking is possible. (our experience from other data sets).

>According to the authors, dispersion curves were estimated using the software of Ryberg et al.,
>2021a. The authors should explain any difference in obtaining the dispersion curve when using the
>acausal or causal parts of EGFs or indicate the validity of using the stack of both sides.

For picking a dispersion curve we used the single-sided part of the EGF. Picking a potential dispersion curve was done the following way: If there was a clear arrival at the frequency of with the highest noise energy (typically between 2-5 Hz) we started picking from that point "extending" the dispersion curve to higher and lower frequency as long as we could observe coherent (continuously connected branch of the dispersion curve). If such a starting point was not present, we did not pick any dispersion curve. The manuscript was modified to make this approach more clear.

>Figure 3 shows an erratic dispersion curve with values between 1.5 and 2.0 km / s. I consider all
>dispersion curves should be displayed and discuss the bandwidth filter effect of the filter.

Given the huge number of dispersion curves, we decided to show only a few examples. The effect of the filter bandwidth effect has been studied intensively and published. We used the values of 0.25

and 3.15 for band and beta (Dziewonski et al., 1969), representing the classical compromise between frequency and time resolution). We used the instantaneous frequency instead of nominal filter frequency to avoid strong "frequency" leakage. We modified Fig. 3 and extended the manuscript accordingly.

>The authors indicate that they resolved 30 frequencies between 1.2 and 20 Hz; if so, it is necessary
>to show the number of velocity values (or arrival times) resolved in each frequency.

Fig. 5 (top) already shows the number of picks (==velocity values) for every frequency!

>How did they go from group velocity to phase velocity to get a Vs model?

We did not use phase velocities for inversion! Only group velocity picks contributed to the inversion.

>Since the 3D inversion model shows a series of contacts with significant velocity (or resistivity)
>contrasts, as indicated by the sections in Figure 10, it is rare that the EGFs in Figure 2 are so
>uniform and do not present some discussion indicating a change in velocity. Consequently, the
>authors should show some virtual source gathers produced along with the seismic stations that
>coincide with Figure 10a sections. These sections let to see that the correlations capture these
>property contrasts.

You observation is right, the example dispersion curve is rather "smooth". We modified Fig. 2 by adding a trace/dispersion curve for a station pair crossing one of the recovered strong (low) velocity anomaly. There the dispersion curve, being continuous (see picking procedure described above), shows a highly dispersive surface wave trace.

---

## Author Comment (AC2)

This is a very well written and comprehensive description of a passive seismic and airborne electromagnetic survey of a zone of greisen mineralisation in Germany. The descriptions of the geological setting, the methods used to acquire the data are complete and accurate. I do not have the background to comment on the processing of the data and I hope this will be done by another more qualified reviewer. I will instead focus on the interpretation of the results, and here I have many questions.

The first question is whether the seismic velocities of the rock types being imaged are sufficiently different that they can be distinguished from one another. And more specifically, can the mineralised greisens be distinguished from surrounding unmineralized rocks? In the manuscript there is considerable discussion about anomalies, both seismic and electromagnetic, and whether they can be related to real geological features. I am not entirely convinced that this has been done. The rocks in the region are all quartzo-feldspathic (mica schists, felsic gneisses and granitoids) and, from literature data, strong differences in velocity are not to be expected. The authors refer to data from Müller-Huber and Börner collected from a near-by area and conclude that greisens might be seismically faster than the surrounding rocks. Yet Müller-Huber and Börner state 'bulk density, however, is critically influenced by porosity and is therefore not suitable to distinguish the Austrian greisen rocks from the surrounding two-mica granites, 3 despite the greisens' comparably high grain density (mean: 2.74 g/cm³). Their higher porosity (mean: 5.7%) also results in lower elastic wave velocities (mostly < 2900 m/s)." The last sentence suggests that the more porous greisens might slower, not faster, than surrounding rocks, but there is not enough information to decide whether this difference is significant.

**We have added petrophysical data (both density data and neutron gamma data as a proxy for porosity) and a paragraph in the methods section of the manuscript to address these concerns. The data show that there is a density contrast between the the quartzo-feldspathic units two-mica schist and muscovite gneiss and that the greisen has a higher density compared to the host granite but a similar porosity.**

The authors have identified anomalies in both the seismic and electromagnetic data and they combine the two using an interesting clustering approach. They derive 9 clusters, which they relate to geological features. Unfortunately, I also found these results to be rather unconvincing.

- The anomaly SA1 is clearly expressed in the seismic models, and in section 3.1 (Ambient noise inversion results) it is related to a contact between a "two-mica schist (high velocities) in the NW and more quartzite-rich mica schist and gneiss (lower velocities)". This contact is not shown on the geological map but there seem to be some discrepancies between the orientation of this anomaly (ca. 045°) and the principal geological structures (ca. 030° to 010°, and ca. 090° south of the Greifenbach fault).

**The reviewer is correct about the slight discrepancy between the orientation of this anomaly and the mapped contact. We have re-evaluated our interpretation, now relating the seismic anomaly SA1 to the transition of hornfels two-mica schist within the contact metamorphic aureole of the granite (high velocities) in the NW and mica schist / muscovite gneiss (lower velocities) due SE. The boundary of the metamorphic aureole of the granite has been added to the geological map, and the text has been modified accordingly.**

- In the authors' discussion of the clusters (section 3.3 "Integration and geological significance"), instead of associating the anomaly SA1 with a change in lithology of the metasedimentary units of

the basement, this cluster is attributed to the Quaternary cover. Given that the anomaly aligns with topographic features (see line 221), the latter interpretation seems more likely.

**This is our mistake. The text should read and has been modified to "Cluster 3 corresponds with the resistivity anomaly EA1, being mostly confined to the surface on both sides of the Greifenbach fault and may therefore represent the Quaternary cover."**

- the contact between the granite intrusion and surrounding metamorphic rocks, where there might be a seismic contrast, is poorly resolved. The position of the contact inferred from mapping and drilling cuts obliquely across the boundary between high and low-velocity zones in Figure 10.

**That's a very good point. Although we expect a velocity contrast at the granite-mica schist interface due to differences in density of these respective units, the seismic data only broadly images this contact, and we observe both high- and low-velocity zones cutting obliquely across this lithological boundary. Thus, we argue that the observed velocity distribution has to be related to factors that are superimposed on the primary lithology, such as weathering, alteration and/or mineralization. Text has been amended to clarify this point.**

- the greisens are related to cluster 8, which is described as an anomaly "at depth having a width of up to 750 m, a length of 1350 m and a thickness of 200 m". This anomaly broadly coincides with most of the "greisen markers" shown in Figure 9 and this may be a useful result, but the size of the anomaly and its boundaries are not well constrained.

**The clusters do not correspond to anomalies per se, but rather define domains of similar properties in two-parameter space. We used k-means as a segmentation method, which is an algorithm that has no notion of outliers, so all points are assigned to a cluster. As mentioned in the manuscript, the lateral dimension of the clusters is thus influenced to a certain degree by the chosen classification parameters as well as the model boundaries and the inherent smoothness of the 3D inversion models. However, as discussed in the text, the mean shear velocities and resistivities of this cluster contrast with those of the classes representing the surrounding non-greisen rocks, so we consider them to be geologically meaningful. Also, cluster 8 forms a spatially coherent volume with a similar dimension as greisen bodies known from the Ehrenfriedersdorf area at the Sauberg and Vierung prospects (Brosig et al, 2020).**

- the seismic study has picked out the Greifenbach Fault, but only in the horizontal slice near the surface where it matches the location defined by the geological mapping and drilling. The fault is not evident at depth in the vertical slices – Figs. 7 and 10.

**In the text, we describe this anomaly as SA2, extending from the surface to an elevation of ca. 500 m, i.e. to a maximum depth of ca. 100 m. In the text, we also say "It also marks the trend of the Greifenbach stream and associated Quaternary deposits.", by which we meant that the seismic data only indirectly maps this fault by its near-surface expression, i.e. the thickened Quaternary cover, which is evident in the geological map (Fig. 1). Text has been modified for clarity.**

In the introduction it is said that the study demonstrates the great potential of the cost-efficient and low-impact ambient noise technology for mineral exploration. How valid is this statement?  It is true that an anomaly that might correspond to the greisens has been identified, but the distribution and margins of the zone are very poorly described.

**See above for reply regarding the constraints of cluster boundaries.**

- The anomaly SA1 seems better related to Quaternary deposits and, if so, has little relevance to the primary lithologies.

**We fail to see how the anomaly SA1, which has a continuity down to ca. 400 m depth and a trend that does not coincide with the Quaternary cover shown in Fig. 1 can be related to Quaternary deposits.**

- The Greifenbach fault is imaged close to the surface but not at depth. It is difficult to see how these results could be a much help in mineral exploration.

**See above for reply on the interpretation of cluster 3.**

What could be done to remove some of these uncertainties? It would be useful for the reader to have better information about the seismic velocities of the different lithologies. The results of Müller-Huber and Börner could be summarised in a table, giving the mean, the range and the uncertainties of the data. This information would help estimate whether the velocity of a two-mica schist is indeed significantly higher than that of quartzite-rich mica schist and gneiss (line 218). In addition, the uncertainty concerning the velocity of the greisen should be resolved – does it have higher porosity that would decrease its velocity below that of the surrounding rocks, as suggested by Müller-Huber and Börner? Would this difference be large enough that the greisens could be imaged in a passive seismic survey? In the present manuscript, this has not been demonstrated.

**See above for reply concerning the petrophysical constraints. In light of the fact that we added new petrophysical data from the study area to the manuscript, we decided not to present a detailed summary of the data from Müller-Huber and Börner (2017) in a table. The presented data in Müller-Huber and Börner (2017) are for rocks from the Altenberg area. While these rocks are of similar age and composition, which warrants a comparison with the rocks of our study area, we feel that including a table with data from that paper would cause confusion. Müller-Huber and Börner do not report petrophysical data for two-mica schist or quartzite-rich mica mica schist and gneiss.**

It would also be useful to superimpose the seismic results on the geological map, as I have tried to do in the attached document.

**We have added the geological boundaries to the horizontal sections (Fig. 11).**

If this type of information could be provided and the probable limitations of the method were discussed, the manuscript should be suitable for publication.

Some minor points

Why wasn't the Quaternary layer imaged with ANSWT? Determination of the thickness and distribution of this cover sequence would be useful information for mineral exploration companies.

**The ANSWT approach cannot be reliably applied to our data because the data come from surface waves with a limited frequency range, i.e. between 1.2–20 Hz, with lower noise in the range between 2 and 10 Hz. Given the high-frequency limit of the picked dispersion curves this means that depths shallower than10–30 m will be poorly resolved. On the other hand, airborne EM data has been used successfully to map aquifer systems and depth to bedrock (e.g., Knight et al., 2018; Christensen et al., 2015). In our study, EA1 is interpreted to correspond well to the mapped extent of Quaternary sediments and cover thickness extracted from drill cores.**

The profile labelled A-A' in the geological map (Fig 1) is oriented NW-SE but in Figure 7, the one labelled A-A' is oriented NE-SW. Please re-label the lines to eliminate this source of confusion.

**Figure and corresponding caption have been modified to eliminate this source of confusion.**

**References:**
**Christensen, C.W., Pfaffhuber, A.A., Anschütz, H., and Smaavik, T.F., 2015, Combining airborne electromagnetic and geotechnical data for automated depth to bedrock tracking: Journal of Applied Geophysics, v. 119, p. 178–191, doi:10.1016/j.jappgeo.2015.05.008.**

**Knight, R., Smith, R., Asch, T., Abraham, J., Cannia, J., Viezzoli, A., and Fogg, G., 2018, Mapping Aquifer Systems with Airborne Electromagnetics in the Central Valley of California: Groundwater, v. 56, p. 893–908, doi:10.1111/gwat.12656.**

---

## Author Response (AR2)

**Comments to the author by Topical Editor Juan Alcalde**
The manuscript presents a very good case study to explore the potential of combined non-invasive geophysical methods for mineral exploration. I would like to congratulate the authors because they have replied thoroughly to the minor revisions proposed by the reviewers, and the paper is almost ready to be published.

There is one last thing that I would like the authors to comment. The new added paragraph on the petrophysical properties of the study area really helps to provide an idea on the differences in properties between to be expected. However, when I observe the sections across the geophysical models (fig. 12), it is clear that the resistivity data provide a stronger constraint to the corresponding classification (12.c). This is also observed in fig. 13.a, which shows that the differences in mean values between shear velocities are smaller than in mean resistivity values. I think that the authors should comment on the influence that the two datasets impose in the final classification and to be more explicit in the text about why the combination of the two techniques provides a more robust understanding of the subsurface architecture of the study area.

**We thank the reviewer for his question and welcome the opportunity to clarify this aspect. We agree that the resistivity anomalies, e.g., the gently NW dipping low-resistivity zones (Fig. 12, rows 1 and 4) impose a strong influence on the classification, but the sections in figure 12 also clearly show the influence of the shear velocity distribution, most prominently seen as vertical contacts in sections of row 1, 3, 6 and 7 both in the shear velocity and in the corresponding classification. We have added an explanation under 3.3 and added row labels to figure 12 to guide the reader to these features. The apparently smaller differences in the mean shear velocities compared to the mean resistivity values in Figure 13A is explained by the fact that the datasets, whose original values exhibit contrasting ranges (1.4–3.0 km/s vs. 1.0–4.0 log Ωm), are plotted on one and the same y-axis. Regarding the explicitness of the integrative approach, we have added another note on this in the conclusion, but the benefits of the integration of two geophysical parameters have also been highlighted in several other places throughout the text (abstract, discussion).**